# The Cost of Energy-Efficiency in Digital Hardware: The Trade-Off between Energy Dissipation, Energy–Delay Product and Reliability in Electronic, Magnetic and Optical Binary Switches

Rahnuma Rahman and Supriyo Bandyopadhyay *

Department of Electrical and Computer Engineering, Virginia Commonwealth University, Richmond, VA 23284, USA; rahmanr3@vcu.edu
* Correspondence: sbandy@vcu.edu

**Featured Application: This work has applications in the benchmarking of binary switches for energy-efficient nanoelectronics.**

**Abstract:** Binary switches, which are the primitive units of all digital computing and information processing hardware, are usually benchmarked on the basis of their 'energy–delay product', which is the product of the energy dissipated in completing the switching action and the time it takes to complete that action. The lower the energy–delay product, the better the switch (supposedly). This approach ignores the fact that lower energy dissipation and faster switching usually come at the cost of poorer reliability (i.e., a higher switching error rate) and hence the energy–delay product alone cannot be a good metric for benchmarking switches. Here, we show the trade-off between energy dissipation, energy–delay product and error–probability for an electronic switch (a metal oxide semiconductor field effect transistor), a magnetic switch (a magnetic tunnel junction switched with spin transfer torque) and an optical switch (bistable non-linear mirror). As expected, reducing energy dissipation and/or energy–delay product generally results in increased switching error probability and reduced reliability.

**Keywords:** binary switches; benchmarking; energy–delay product; reliability

## 1. Introduction

The primitive element of all digital circuits (i.e., for computing, signal processing, etc.) is a "binary switch" which has two stable states encoding the binary bits 0 and 1. Computing and digital signal processing tasks are carried out by flipping such switches back and forth between the two states. As a result, for a given algorithm and a given computing architecture, the energy cost and speed of a digital computational task are determined by the energy dissipation and the switching delay of the switches. Therefore, it has become common practice to benchmark digital switches on the basis of their 'energy–delay product', which is the product of the energy dissipated during switching and the switching time [1].

This approach, unfortunately, ignores the fact that usually the less energy we dissipate (or the faster we try to switch), the more error-prone the switch becomes. As a result, any saving in energy or computational time gained by employing switches with a lower energy–delay product may be offset by the additional resources that would be needed for error correction. In this paper, we show the direct relation between energy dissipation and error-resilience with three examples, a field effect transistor, a nanomagnetic switch flipped with current induced spin–transfer–torque [2] and a non-linear mirror.

## 2. Field-Effect-Transistor Switch

A metal–oxide–semiconductor–field–effect–transistor (MOSFET) is the archetypal (electronic) binary switch that encodes the bits 0 and 1 in its two conductance states, high (ON) and low (OFF). In the ON state, charges flood into the channel providing a conduction path between the source and the drain to turn the transistor on. In the OFF state, these charges are driven out of the channel in order to disrupt the conduction path and turn the transistor off. Therefore, the two states are ultimately encoded in two *different amounts of charge* ($Q_1$ and $Q_2$) in the channel. The switching action changes the amount of charge from $Q_1$ to $Q_2$, or vice versa, resulting in the (time-averaged) flow of a current:

$$I = |Q_1 - Q_2|/\Delta t = \Delta Q/\Delta t \tag{1}$$

where $\Delta t$ is the amount of time it takes for the channel charge to change from $Q_1$ to $Q_2$ (or vice versa). This current will cause energy dissipation of the amount:

$$E_d = I^2 R \Delta t = (\Delta Q/\Delta t) I R \Delta t = \Delta Q I R = \Delta Q \Delta V \tag{2}$$

where $R$ is the resistance in the path of the current and $\Delta V = IR$. We can think of $\Delta V$ as the amount of voltage needed to be imposed at the transistor's gate in order to change the charge in the channel by the amount $\Delta Q$. Note that the energy dissipation given in Equation (2) is *not* independent of the switching time, because $\Delta V$ depends on the switching time for a fixed $\Delta Q$ and $R$ ($\Delta V = \Delta Q R/\Delta t$). We can rewrite the energy dissipation in Equation (2) as $E_d = (\Delta Q)^2 R/\Delta t$, which clearly shows that for a fixed $\Delta Q$ and $R$, we will dissipate *more* energy if we switch *faster* (smaller $\Delta t$). Therefore, a more meaningful quantity to benchmark energy-efficiency is the *energy–delay product* which is $E_d \Delta t = (\Delta Q)^2 R$. For a fixed $R$, we can reduce this quantity by reducing $\Delta Q$, but that increasingly blurs the distinction between $Q_1$ and $Q_2$ and thereby impairs our ability to distinguish between bits 0 and 1. If $\Delta Q$ is too small, then thermal generation and recombination can randomly change the amount of charge in the channel by an amount comparable to $\Delta Q$, causing random switching. Therefore, a larger $\Delta Q$ translates to both stronger error-resilience and better reliability. This makes it obvious that there is a direct relation between reliability and energy–delay product; if we reduce the energy dissipation or energy-delay product by reducing $\Delta Q$, then we will invariably make the switch less reliable.

We can make this argument a little more precise by noting that $\Delta Q = C_g \Delta V$, where $C_g$ is the gate capacitance. The thermal voltage fluctuation at the gate terminal is given by $\sqrt{kT/C_g}$, where $kT$ is the thermal energy [3]. Hence, the thermal charge fluctuation in the channel is:

$$\Delta Q|_{\text{fluctuation}} = \sqrt{C_g kT} \tag{3}$$

This quantity must be much smaller than the $\Delta Q$ one needs to switch the conductance state of the transistor, and hence $\Delta Q|_{\text{fluctuation}} << \Delta Q$. Let us define a quantity $\eta$ such that $\eta = \Delta Q/(\Delta Q|_{\text{fluctuation}})$. Clearly, $\eta$ is a measure of the 'switching reliability'; the larger its value, the more reliable is the switch. From Equation (2), we can now obtain:

$$E_d = \frac{(\Delta Q)^2}{C_g} = \frac{\eta^2 (\Delta Q|_{\text{fluctuation}})^2}{C_g} = \eta^2 kT$$
$$E_d \Delta t = (\Delta Q)^2 R = \eta^2 kTRC_g \tag{4}$$

which immediately shows that we have to tolerate more energy dissipation, $E_d$, and larger energy–delay product, $E_d \Delta t$, if we desire more reliability (i.e., a larger $\eta$) [4].

In some specific cases, such as a field–effect–transistor, we may be able to derive a relation between the energy dissipation/energy–delay product and the error probability. Consider the conduction-band diagram in the channel of an *n*-channel field–effect–transistor along the direction of drain current flow as shown in Figure 1. In the OFF state, there is a potential barrier at the source–channel junction which prevents electrons in the source contact from entering the channel and turning the transistor ON. This barrier has to be

lowered by the applied gate potential $\Delta V$ in order to allow electrons to enter the channel when the transistor has to be turned ON. Therefore, this barrier should be approximately equal to the quantity $q\Delta V$. It is clear then that the transistor can spontaneously turn ON *while in the non-conducting state* (causing a switching error that results in a bit error), if electrons can enter the channel from the source by thermionic emission over the barrier. The probability of entering the channel in this fashion, which is roughly $e^{-(q\Delta V)/kT}$, is then the switching error probability $p$. From Equation (2), we then get that the energy dissipation can be written as:

$$E_d = \Delta Q \Delta V = C_g(\Delta V)^2 = C_g[(kT/q)\ln(1/p)]^2 \tag{5}$$

and the energy–delay product can be written as:

$$E_d\Delta t = (RC_g)C_g(\Delta V)^2 = \tau C_g[(kT/q)\ln(1/p)]^2 \tag{6}$$

where $\tau = RC_g$ is the gate charging time. Equations (5) and (6) show the direct dependences of the energy dissipation and energy–delay product on the error-probability $p$. These two equations clearly show that lower energy dissipation or lower energy–delay product are associated with higher switching error probability in a transistor switch.

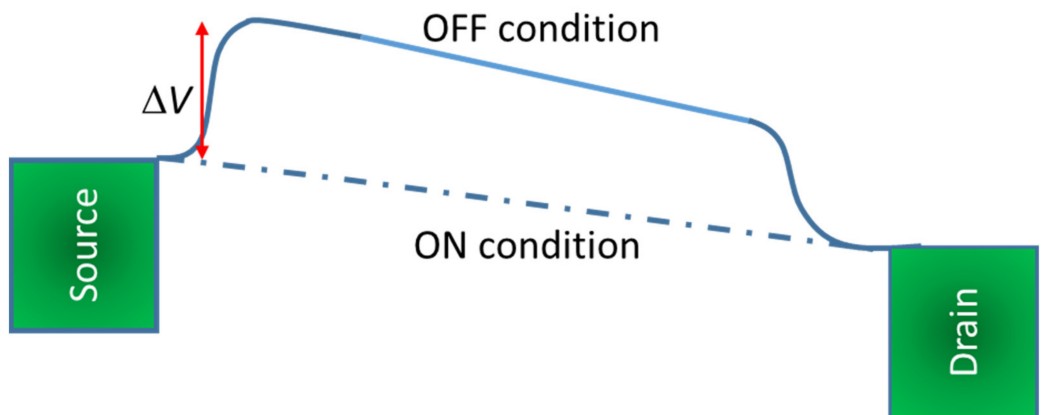

**Figure 1.** Conduction band profile along the channel of a field effect transistor in the OFF state (solid line) and ON state (broken line).

To provide an estimate of how much energy we must dissipate to maintain any semblance of reliability, we will assume that we can tolerate a maximum error probability of $10^{-15}$ [5]. In modern day FINFETs, the gate capacitance may vary between 10 and 20 aF for a 0.5 μm wide gate [6], and $\tau$ is on the order of 100 ps. Using the values in Equations (5) and (6), we get that the minimum energy dissipation and the minimum energy–delay product that we can expect in this type of FET device, while maintaining minimum acceptable reliability, are ~10 aJ and $10^{-27}$ J-s, respectively.

## 3. Nanomagnetic Switches

Next, we consider a magnetic switch. We first point out that there are two types of switching errors: "unintentional switching" (i.e., switching that takes place when it is *not* desired) and "failure to switch" (i.e., switching that does *not* take place when it is desired). The previous example with the field–effect–transistor is of the first type and relates to errors caused by unintentional switching. Magnetic switches, fashioned out of nanomagnets (e.g., magnetic tunnel junctions), can also experience unintentional switching due to stray magnetic fields, thermal agitations, etc. The relationship between the energy dissipation

and error probability for unintentional switching was derived in ref. [7], which deduced that the minimum energy dissipation is related to the error probability according to:

$$E_d^{\min} = 2kT \ln\left[-\frac{T_c}{\tau_0 \ln(1-p)}\right] \tag{7}$$

where $T_c$ is the clock period and $\tau_0$ is the inverse of the attempt frequency (which is in the range of 1 ps–1 ns) [8].

To maintain $p = 10^{-15}$ at a clock frequency of 1 GHz, the minimum energy dissipation would be 0.14–0.17 aJ according to Equation (7). We caution that this is an overly optimistic estimate, since it assumes that the minimum energy dissipation needed to switch is the anisotropy energy barrier within the nanomagnet and no consideration has been made of additional energy losses due to Gilbert damping, etc.

Next, we address "failure to switch" in a magnetic switch (i.e., switching not taking place when switching is desired and a switching stimulus is provided). Unlike the electronic switch, this case is not amenable to any analytical treatment and hence we will resort to simulations, using a specific example.

A bistable nanomagnetic switch can be fashioned out of a ferromagnetic elliptical disk where, because of the elliptical shape, the magnetization can point only along the major axis, either pointing to the left or to the right (as shown in Figure 2a). This type of nanomagnet is said to possess in-plane magnetic anisotropy (IPA). In thinner nanomagnets, the surface anisotropy may be dominant and the magnetization can point perpendicular to the surface (either up or down). This type of nanomagnet is said to possess perpendicular magnetic anisotropy (PMA) (Figure 2b). Either type makes a binary switch if we encode the bit information in the magnetization orientation which can point in just two directions. In this paper, we will consider only the IPA nanomagnetic switch, although the results will apply equally to PMA nanomagnets.

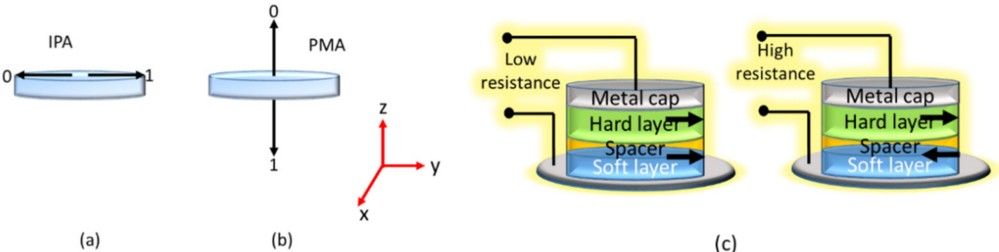

**Figure 2.** A nanomagnet shaped like an elliptical disk has two stable magnetization orientations which can encode the binary bits 0 and 1. (**a**) In-plane magnetic anisotropy and (**b**) perpendicular magnetic anisotropy. (**c**) A magnetic tunnel junction (MTJ) showing the high (OFF) and low (ON) resistance states.

The IPA nanomagnet can be vertically integrated as the "soft" layer into a three-layer stack consisting of a "hard" ferromagnetic layer and an insulating (non-magnetic) spacer to form a magnetic tunnel junction (MTJ) (as shown in Figure 2c). The hard layer is permanently magnetized in one of its two stable directions. When the soft layer's magnetization is parallel to that of the hard layer, the MTJ resistance (measured between the two ferromagnetic layers) is low. If the two magnetizations are antiparallel, the resistance is high. Thus, the MTJ acts as a binary switch, much like the transistor, whose two resistance states (i.e., high and low) encode the binary bits 0 and 1. The difference between the transistor and the MTJ is that the former is *volatile* (since charges leak out when the device is powered off), while the MTJ is *non-volatile* (since the bit information is encoded in magnetization (spins) and not charge).

In order to make the magnetizations of the hard and soft layers mutually parallel (ON state), we can employ spin–transfer–torque [2]. To do this, we apply a voltage across the MTJ with the negative polarity of the battery connected to the hard layer. This will inject

spin-polarized electrons from the hard layer into the soft layer, whose spins are mostly aligned along the magnetization orientation of the hard layer. These injected electrons will transfer their spin angular momenta to the resident electrons in the soft layer, whose spins will then gradually turn in the direction of the injected spins and will magnetize the soft layer in a direction parallel to the magnetization of the hard layer. This is how the MTJ is turned "on". In order to turn it "off", we will reverse the polarity of the battery. This will inject electrons from the soft layer into the hard layer. However, because of spin-dependent tunneling through the spacer, those electrons whose spin polarizations are parallel to the magnetization of the hard layer will be preferentially injected. As these spins exit the soft layer their population is quickly depleted, leaving the opposite spins as the majority in the soft layer. That aligns the magnetization of the soft layer antiparallel to that of the hard layer. When that happens, the MTJ turns off.

For any given magnitude of injected current (with a given degree of spin polarization $\zeta$), we can calculate the switching error probability (at room temperature) associated with spin–transfer–torque switching of an MTJ by carrying out **Landau–Lifshitz–Gilbert–Langevin** simulation (also known as stochastic Landau–Lifshitz–Gilbert (s–LLG) simulation). To do this, we solve the following equation:

$$
\begin{aligned}
\frac{d\vec{m}(t)}{dt} &= -\gamma\vec{m}(t) \times \vec{H}_{\text{total}}(t) + \alpha\left(\vec{m}(t) \times \frac{d\vec{m}(t)}{dt}\right) \\
&\quad + a\vec{m}(t) \times \left(\frac{\zeta\vec{I}_s(t)\mu_B}{qM_s\Omega} \times \vec{m}(t)\right) + b\frac{\zeta\vec{I}_s(t)\mu_B}{qM_s\Omega} \times \vec{m}(t)
\end{aligned}
$$

where

$$
\begin{aligned}
\hat{m}(t) &= m_x(t)\hat{x} + m_y(t)\hat{y} + m_z(t)\hat{z} \quad \left[m_x^2(t) + m_y^2(t) + m_z^2(t) = 1\right] \\
\vec{H}_{\text{total}} &= \vec{H}_{\text{demag}} + \vec{H}_{\text{thermal}} \\
\vec{H}_{\text{demag}} &= -M_sN_{d-xx}m_x(t)\hat{x} - M_sN_{d-yy}m_y(t)\hat{y} - M_sN_{d-zz}m_z(t)\hat{z} \\
\vec{H}_{\text{thermal}} &= \sqrt{\frac{2\alpha kT}{\gamma(1+\alpha^2)\mu_0 M_s\Omega(\Delta t)}}\left[G_{(0,1)}^x(t)\hat{x} + G_{(0,1)}^y(t)\hat{y} + G_{(0,1)}^z(t)\hat{z}\right]
\end{aligned}
$$

(8)

The last term in the right-hand side of Equation (8) is the field-like spin transfer torque exerted by the injected current $I_s$ and the second to last term is the Slonczewski torque exerted by the same current. The coefficients $a$ and $b$ depend on device configurations and, following [9], we will use the values $a$ = 1, $b$ = 0.3. Here, $\vec{m}(t)$ is the time-varying magnetization vector in the soft layer normalized to unity, $m_x(t)$, $m_y(t)$, and $m_z(t)$ are its time-varying components along the x-, y-, and z-axis, respectively (see Figure 2 for the Cartesian axes), $\vec{H}_{\text{demag}}$ is the demagnetizing field in the soft layer due to its elliptical shape, and $\vec{H}_{\text{thermal}}$ is the random magnetic field due to thermal noise [10]. The different parameters in Equation (3) are: $\gamma = 2\mu_B\mu_0/\hbar$ (gyromagnetic ratio), $\alpha$ is the Gilbert damping constant, $\mu_0$ is the magnetic permeability of free space, $M_s$ is the saturation magnetization of the cobalt soft layer, $kT$ is the thermal energy, $\Omega$ is the volume of the soft layer which is given by $\Omega = (\pi/4)a_1 a_2 a_3$ [$a_1$ = major axis, $a_2$ = minor axis and $a_3$ = thickness], $\Delta t$ is the time step used in the simulation, and $G_{(0,1)}^x(t)$, $G_{(0,1)}^y(t)$ and $G_{(0,1)}^z(t)$ are three uncorrelated Gaussians with zero mean and unit standard deviation [10]. The quantities $N_{d-xx}, N_{d-yy}, N_{d-zz}$ $\left[N_{d-xx} + N_{d-yy} + N_{d-zz} = 1\right]$ are calculated from the dimensions of the elliptical soft layer following the prescription of ref. [11]. The nanomagnet soft layer is assumed to be made of cobalt with saturation magnetization $M_s$ = 8 × 10$^5$ A/m and $\alpha$ = 0.01. Its major axis = 800 nm, minor axis = 700 nm and thickness = 2.2 nm. We assume that the spin polarization in the injected current is $\zeta$, which we take to be 30%. The spin current is given by $\zeta\vec{I}_s(t) = \zeta\left|\vec{I}_s(t)\right|\hat{y}$.

Using the vector identity $\vec{a} \times \left( \vec{b} \times \vec{c} \right) = \vec{b} \left( \vec{a} \bullet \vec{c} \right) - \vec{c} \left( \vec{a} \bullet \vec{b} \right)$, we can recast the vector equation in Equation (8) as [12]:

$$
\begin{aligned}
(1+\alpha^2)\frac{d\vec{m}(t)}{dt} &= -\gamma\left( \vec{m}(t) \times \vec{H}_{\text{total}}(t) \right) - \gamma\alpha\left[ \vec{m}(t)\left( \vec{m}(t)\bullet\vec{H}_{\text{total}}(t) \right) - \vec{H}_{\text{total}}(t) \right] \\
&\quad -(\alpha a - b)\left( \frac{\xi \vec{I}_s(t)\mu_B}{qM_s\Omega} \times \vec{m}(t) \right) + (a+\alpha b)\frac{\xi I_s(t)\mu_B}{qM_s\Omega}\left[ \hat{y} - \vec{m}(t)m_y(t) \right]
\end{aligned}
\tag{9}
$$

This vector equation can be recast as three coupled scalar equations in the three Cartesian components of the magnetization vector [12]:

$$
\begin{aligned}
(1+\alpha^2)\frac{dm_x(t)}{dt} &= -\gamma\left[ m_y(t)H_z(t) - m_z(t)H_y(t) \right] - \alpha\gamma\left[ m_x(t)\left[ m_x(t)H_x(t) + m_y(t)H_y(t) + m_z(t)H_z(t) \right] - H_x(t) \right] \\
&\quad -(a\alpha - b)\frac{\xi I_s(t)m_z(t)\mu_B}{qM_s\Omega} - (a+\alpha b)\frac{\xi I_s(t)m_x(t)m_y(t)\mu_B}{qM_s\Omega} \\
(1+\alpha^2)\frac{dm_y(t)}{dt} &= -\gamma\left[ m_z(t)H_x(t) - m_x(t)H_z(t) \right] - \alpha\gamma\left[ m_y(t)\left[ m_x(t)H_x(t) + m_y(t)H_y(t) + m_z(t)H_z(t) \right] - H_y(t) \right] \\
&\quad +(a+\alpha b)\frac{\xi I_s(t)\mu_B}{qM_s\Omega}\left( 1 - m_y^2(t) \right) \\
(1+\alpha^2)\frac{dm_z(t)}{dt} &= -\gamma\left[ m_x(t)H_y(t) - m_y(t)H_x(t) \right] - \alpha\gamma\left[ m_z(t)\left[ m_x(t)H_x(t) + m_y(t)H_y(t) + m_z(t)H_z(t) \right] - H_z(t) \right] \\
&\quad +(a\alpha - b)\frac{\xi I_s(t)m_x(t)\mu_B}{qM_s\Omega} - (a+\alpha b)\frac{\xi I_s(t)m_z(t)m_y(t)\mu_B}{qM_s\Omega}
\end{aligned}
\tag{10}
$$

where

$$
\begin{aligned}
H_x &= -M_s N_{d-xx} m_x(t) + \sqrt{\frac{2\alpha kT}{\gamma(1+\alpha^2)\mu_0 M_s\Omega(\Delta t)}} G^x_{(0,1)}(t) \\
H_y &= -M_s N_{d-yy} m_y(t) + \sqrt{\frac{2\alpha kT}{\gamma(1+\alpha^2)\mu_0 M_s\Omega(\Delta t)}} G^y_{(0,1)}(t) \\
H_z &= -M_s N_{d-zz} m_z(t) + \sqrt{\frac{2\alpha kT}{\gamma(1+\alpha^2)\mu_0 M_s\Omega(\Delta t)}} G^z_{(0,1)}(t)
\end{aligned}
$$

In our s–LLG simulations, we considered six different switching currents of 0.5, 1.0, 5.0, 10.0, 15.0 and 20.0, corresponding to current densities of $1.14 \times 10^9$ A/m$^2$, $2.27 \times 10^9$ A/m$^2$, $1.14 \times 10^{10}$ A/m$^2$, $2.27 \times 10^{10}$ A/m$^2$, $3.41 \times 10^{10}$ A/m$^2$ and $4.55 \times 10^{10}$ A/m$^2$, respectively. We generated 1000 switching trajectories for each current by solving Equation (10). We start with the initial condition $m_x(t=0) = 0.1, m_y(t=0) = -0.99, m_z(t=0) = 0.1$ and ran each trajectory for 20 ns with a time step of 0.1 ps. After 20 ns, each trajectory ended with a value of $m_y$ either close to +1 (switching success) or −1 (switching failure). The error probability is the fraction of trajectories that resulted in failure.

In Figure 3, we plot the error probability (at room temperature) as a function of the current injected. Keeping in mind that the bulk of the energy dissipated is proportional to the square of the current, we see that the error probability decreases monotonically with increasing current or increasing energy dissipation. This shows that energy efficiency can only be purchased at the cost of reliability. In this respect, the magnetic switch shows the same trait as the electronic switch. In both cases, we have to expend more energy during switching if we wish to increase switching reliability.

In Figure 4, we show the error probability as a function of switching time (pulse width of the injected current) for a fixed magnitude of the current. The current strength chosen for this plot was 10 mA. In this simulation, we turned off the current after different intervals of time and continued the simulation for 20 ns to see whether the value of $m_y$ ended up close to +1 (success) or −1 (failure). Again, the simulation duration of 20 ns was sufficient to ensure that for each simulated trajectory, $m_y$ ended up close to either +1 (success) or −1 (failure) at the end of the simulation. One thousand switching trajectories were generated for each pulse width, and the error probability is the fraction of trajectories that result in failure. We observe that the error probability decreased with increasing current pulse width (longer passage of current, or slower switching), as expected.

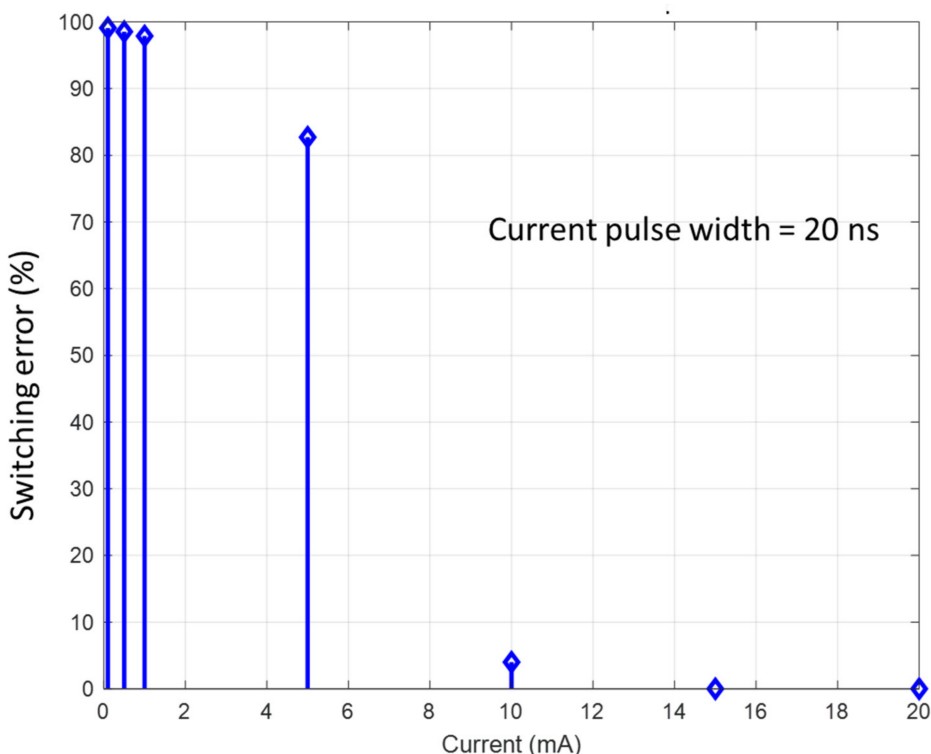

**Figure 3.** Switching error probability as a function of injected current magnitude. The energy dissipated is proportional to the square of the current. The current was kept on for the entire duration of the simulation, which was 20 ns.

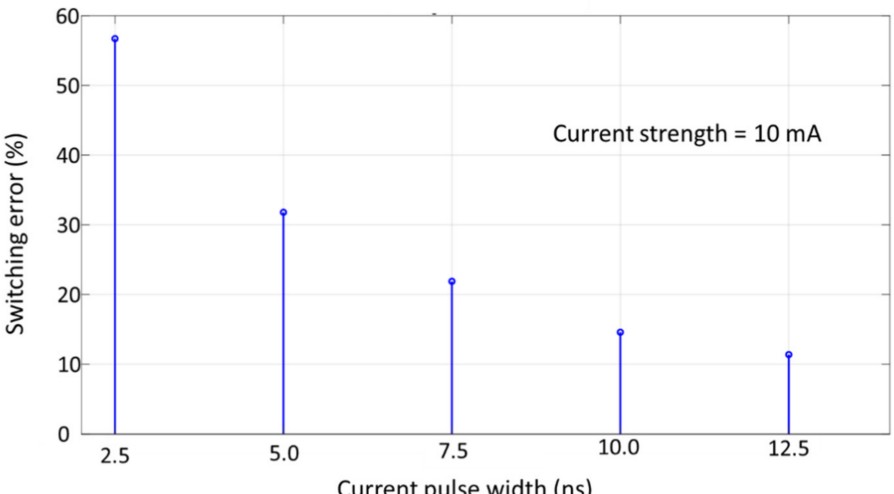

**Figure 4.** Switching error probability as a function of current pulse width (i.e., the duration of spin–transfer–torque). The current strength was kept fixed at 10 mA.

## 4. All-Optical Switches

Optical switches can be switched with a variety of agents such as electric fields, strain, another light beam, etc. A comprehensive study of all switching modalities will be beyond the scope of this article. Hence, we will focus only on all-optical switching [13,14]. We will study the simple case of a non-linear mirror switch shown in Figure 5.

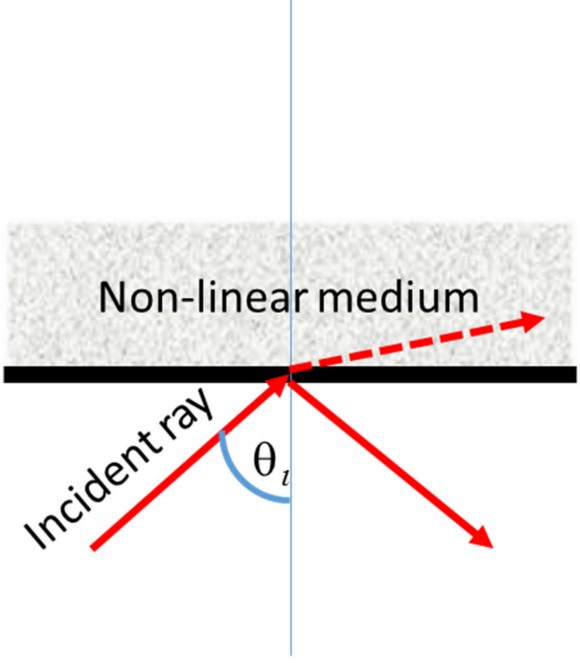

**Figure 5.** A non-linear optical switch.

Consider a ray incident from a linear medium onto a non-linear medium at an angle $\theta_l$. If the angle of incidence exceeds the critical angle for this pair of media, the ray suffers total internal reflection. Otherwise, it is refracted into the non-linear medium. The critical angle is given by the expression:

$$\theta_c = \sin^{-1}\left(\frac{n^{(2)}}{n^{(1)}}\right) = \sin^{-1}\left(\frac{n_0^{(2)} - n_2^{(2)}I}{n^{(1)}}\right), \tag{11}$$

where $n^{(1)}$ is the refractive index of the medium of incidence and $n^{(2)}$ is the refractive index of the (non-linear) medium of refraction $\left(n^{(2)} < n^{(1)}\right)$. Since the latter medium is non-linear, its refractive index depends on the intensity of light incident on it, so that: $n^{(2)} = n_0^{(2)} - n_2^{(2)}I$, where $I$ is the light intensity (related to the energy dissipation in switching). The switch works in the following way. Initially, the ray is incident at an angle less than the critical angle $(\theta_i < \theta_c)$. Then, as the intensity is increased and the critical angle decreases according to Equation (11), at some point the angle of incidence exceeds the critical angle $(\theta_i \geq \theta_c)$ and the ray suffers total internal reflection.

At any incident angle $\theta_i$, the minimum intensity needed to switch is given by the relation:

$$I_{\min} = \frac{n_0^{(2)} - n^{(1)}\sin\theta_i}{n_2^{(2)}} \tag{12}$$

The probability of switching will go up rapidly with $\theta_i$ exceeding $\theta_c$, (i.e., with the difference $\theta_i - \theta_c$). Therefore, the probability will go up rapidly with increasing value of the quantity $\underbrace{\sin^{-1}\left(\frac{n_0^{(2)} - n_2^{(2)}I_{\min}}{n^{(1)}}\right)}_{\text{fixed}} - \sin^{-1}\left(\frac{n_0^{(2)} - n_2^{(2)}I}{n^{(1)}}\right)$, which shows that the probability of successful switching increases with increasing intensity $I$, or increasing energy dissipation. Thus, there is once again a trade-off between energy dissipation and reliability.

## 5. Conclusions

In this article, we have shown the relationship between energy dissipation, switching delay, and reliability for binary switches used in nanoelectronics, nanomagnetics and even nanophotonics. Typically, energy efficiency and faster switching come at the cost of reduced reliability. Consequently, it is not appropriate to benchmark switching devices only in terms of their energy–delay product, since a lower energy–delay product can always be purchased at the cost of error-resilience. This result should be intuitive, and here we have established that with concrete examples.

This begs the question if there are electronic, magnetic and optical computing and information processing paradigms that can tolerate high error probabilities because they can afford to be more energy-efficient. Boolean logic, which is at the heart of most arithmetic logic units in modern day computers, demands a high degree of reliability [15] and therefore is not likely to be frugal in its use of energy. On the other hand, there are computing paradigms (e.g., neuromorphic, probabilistic, Bayesian, etc.) where the computational activity is often elicited from the collective activity of many devices (switches) working in unison. In those cases, a single device (or few devices) being erratic does not impair overall circuit functionality [16]. Consequently, they can tolerate much higher error probabilities. Hardware platforms for these non-Boolean computing paradigms are therefore likely to be more energy efficient than Boolean logic, which has already motivated a great deal of interest in them [15].

**Author Contributions:** Conceptualization, S.B.; methodology, S.B.; software, R.R.; validation, R.R.; formal analysis, S.B.; investigation, R.R. and S.B.; resources, S.B.; data curation, R.R.; writing—original draft preparation, S.B.; writing—review and editing, S.B.; visualization, R.R.; supervision, S.B.; project administration, S.B.; funding acquisition, S.B. All authors have read and agreed to the published version of the manuscript.

**Funding:** This research was supported by the US National Science Foundation under grant CCF-2001255.

**Institutional Review Board Statement:** Not applicable.

**Informed Consent Statement:** Not applicable.

**Data Availability Statement:** All data available are in the article.

**Conflicts of Interest:** The authors declare no conflict of interest.

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
