# Peer review of "The Cost of Energy-Efficiency in Digital Hardware: The Trade-Off between Energy Dissipation, Energy–Delay Product and Reliability in Electronic, Magnetic and Optical Binary Switches"

_applsci, doi:10.3390/app11125590_

Round 1

Reviewer 1 Report

The authors discuss the interrelation of energy dissipation, energy‐delay product and reliability in electronic and magnetic binary switches regarding costs of energy efficiency for a metal oxide 22 semiconductor field-effect transistor and a magnetic tunnel junction 23 switched with spin-transfer torque. Their research confirms that in general, by reducing energy dissipation and/or energy‐delay‐product, switching error probability increases and the reliability reduces. The research brings about an important topic to understand the energy consumption of electronic devices at the basic level.

The research objective and the methodology used is quite match up. However, the article does not follow a traditional structure: Intro, Method, Results, Discussion, and Conclusion. Indeed, it is not compulsory for the authors to adopt such a structure, but by doing so, the article should be a lot more readable for some readers. Please consider. At least you need to emphasize the methods.

For this kind of study, a literature review should get higher attention than theoretical background (not necessarily regarding the length). Therefore, a literature review on the topic must be enhanced by including more works from previous studies.

The overall level of the article is good; even it is quite simple at this stage. Please show a calculation example to Section 2 to provide a quantitive impression of the model that you need to support your conclusion.

Further, please see the details below.

  • Try to increase colour contrast in Figure 2c,
  • Where is Section 3?

Author Response

We thank the reviewer for the suggestions and address them point-by-point below.

The research objective and the methodology used is quite match up. However, the article does not follow a traditional structure: Intro, Method, Results, Discussion, and Conclusion. Indeed, it is not compulsory for the authors to adopt such a structure, but by doing so, the article should be a lot more readable for some readers. Please consider. At least you need to emphasize the methods.

Response: This structure is present. We do have an Introduction, Results and Discussion, and Conclusion. This is not an experimental paper and hence we do not have a Methods section.

For this kind of study, a literature review should get higher attention than theoretical background (not necessarily regarding the length). Therefore, a literature review on the topic must be enhanced by including more works from previous studies.

Response: We did conduct a literature review, but there is hardly any literature on this subject. Surprisingly, even though the result should be intuitive, there is little in the literature that discusses this topic explicitly.

Please show a calculation example to Section 2 to provide a quantitive impression of the model that you need to support your conclusion.

Response: We have added this numerical example in the revised version.

  • Try to increase colour contrast in Figure 2c,… Response: We have changed the color of the lettering. The actual figure has 300 dpi resolution.
  • Where is Section 3? Response: Section 4 was misnumbered. It should have been Section 3. This has been corrected. Thanks for pointing this out.

Reviewer 2 Report

In this manuscript Authors discuss the trade off between Energy-delay product and the reliability in two fundamental building blocks of binary switches (1) a FET switch and (2) a MTJ. Authors conclude that reducing the energy-delay product (i.e. low energy operation and fast switching) comes at the price of a higher error rate. In the first example (i.e., FET switching), a simple back-of-the-envelope analytical approach based on changing/discharging of the FET channel is employed to prove the concept. In the second example (i.e., MTJ); a computational approach based on LLG model is employed.

Overall, I believe the manuscript is very well written, has a fluent language, and takes a simple approach towards connecting the energy-delay product to the switching reliability. From a more technical point of view, I can say that there is nothing surprising in the findings presented here, mostly expected trends; lower switching energy (i.e., lower power or faster switching) makes the device prone to negative influence from noise. But, this manuscript can serve as a useful pedagogical reference, so I can still see the merit in having it published in a public domain. However, I would like the following point to be addressed very carefully before the publication of the manuscript.

  • In both examples, authors assume that thermal fluctuations are the only source of noise/error. It will be useful to better motivate this point in the introduction part of the manuscript and explain why other source of error are not considered here. For example, in the dense integration of MTJs, a major concern is the crosstalk between the neighboring cells, meaning that the stray field of the magnets used in one MTJ can make unintentional switching in neighboring MTJs. This has been a serious issue for this technology and at the same time a driving force for moving towards anti-ferromagnetic devices with a zero-net magnetization. In this sense, stary field in the form of dipolar interactions are also a major source or error. Similar arguments can be made in the case of FET switches.
  • Another important set of switches that are becoming more and more popular are the optical switches in which the inherently faster speed of light is employed for enabling ultra-fast signal processing. In this technology, everything depends on changing the refractive index of the material using a beam light. That’d be unique if authors can add a session (not necessarily extended), or at least a paragraph, to the revised manuscript and discuss how the energy-delay product and reliability can be related in optical switches. The following article provides a background for optical switching: ACS Photonics 2019 “All-Optical Control of Light in Micro- and Nanophotonics”. Here is also one of the fastest optical switches demonstrated so far: Advanced Material 2018 “Hot-Electron-Assisted Femtosecond All-Optical Modulation in Plasmonics”

Author Response

We thank the reviewer for a careful review and helpful suggestions. A point-by-point response is appended below.

Overall, I believe the manuscript is very well written, has a fluent language, and takes a simple approach towards connecting the energy-delay product to the switching reliability. From a more technical point of view, I can say that there is nothing surprising in the findings presented here, mostly expected trends; lower switching energy (i.e., lower power or faster switching) makes the device prone to negative influence from noise. But, this manuscript can serve as a useful pedagogical reference, so I can still see the merit in having it published in a public domain.

Response: The reviewer is right that the result is intuitive. What we have done is take it beyond intuition and establish it on a firm footing with examples and numerical results.

In both examples, authors assume that thermal fluctuations are the only source of noise/error. It will be useful to better motivate this point in the introduction part of the manuscript and explain why other source of error are not considered here. For example, in the dense integration of MTJs, a major concern is the crosstalk between the neighboring cells, meaning that the stray field of the magnets used in one MTJ can make unintentional switching in neighboring MTJs. This has been a serious issue for this technology and at the same time a driving force for moving towards anti-ferromagnetic devices with a zero-net magnetization. In this sense, stary field in the form of dipolar interactions are also a major source or error. Similar arguments can be made in the case of FET switches.

Response: There are two kinds of errors: “unintentional switching”, i. e. switching taking place when it is NOT desired, and “failure to switch”, i. e. switching NOT taking place when it is desired. We have now made this clear in the paper. The first example in the paper (FET) is of the former type and the second (MTJ) is of the latter type. The reviewer is alluding to errors of the former type. In the context of magnetic devices, our group had earlier examined errors of the former type (caused by stray fields, thermal noise, etc.) and derived an expression relating the minimum energy dissipation to the error probability. We have now reported that relation in this paper and cited our earlier work as a reference.

Another important set of switches that are becoming more and more popular are the optical switches in which the inherently faster speed of light is employed for enabling ultra-fast signal processing. In this technology, everything depends on changing the refractive index of the material using a beam light. That’d be unique if authors can add a session (not necessarily extended), or at least a paragraph, to the revised manuscript and discuss how the energy-delay product and reliability can be related in optical switches. The following article provides a background for optical switching: ACS Photonics 2019 “All-Optical Control of Light in Micro- and Nanophotonics”. Here is also one of the fastest optical switches demonstrated so far: Advanced Material 2018 “Hot-Electron-Assisted Femtosecond All-Optical Modulation in Plasmonics”

Response: This is indeed a very valuable suggestion. The refractive index can be changed in various ways, not just with light, but also electric fields, strain, etc. Since, we cannot address all modalities, we will restrict ourselves to all-optical modulators as in the reference invoked by the referee (M. Taghinejad and W. Cai, All-optical control of light in micro- and nanophotonics, ACS Photonics, 6, 1082-1093 (2019)). Using a very simple non-linear all-optical switch model, we show qualitatively that increasing optical intensity (energy dissipation) will increase the switching probability in this case. This discussion is very qualitative; but going beyond that will be outside the scope of this paper. However, this reinforces the message of the paper by adding one other example, this one a non-linear optical switch. Once again, we show that dissipating more energy increases reliability.

There is a dogma in the device community [see ref. [1]] that switching devices should be benchmarked on the basis of their energy dissipation or energy-delay product. We point out that this is a flawed approach since it does not take into account the issue of reliability. One can always purchase energy efficiency at the cost of error-resilience, but such a strategy can be self-defeating.